# Walking down Skeletal Muscle Lane: From Inflammasome to Disease

**DOI:** 10.3390/cells10113023

**Published:** 2021-11-04

**Authors:** Nicolas Dubuisson, Romain Versele, María A. Davis-López de Carrizosa, Camille M. Selvais, Sonia M. Brichard, Michel Abou-Samra

**Affiliations:** 1Endocrinology, Diabetes and Nutrition Unit, Institute of Experimental and Clinical Research, Medical Sector, Université Catholique de Louvain, 1200 Brussels, Belgium; romain.versele@uclouvain.be (R.V.); mayadavis@us.es (M.A.D.-L.d.C.); camille.selvais@uclouvain.be (C.M.S.); sonia.brichard@uclouvain.be (S.M.B.); michel.abousamra@uclouvain.be (M.A.-S.); 2Neuromuscular Reference Center, Cliniques Universitaires Saint-Luc, Avenue Hippocrate 10, 1200 Brussels, Belgium; 3Departamento de Fisiología, Facultad de Biología, Universidad de Sevilla, 41012 Seville, Spain

**Keywords:** skeletal muscle, NLRP3, inflammasome, pyroptosis, metabolic syndrome, sepsis, critical limb ischemia, mmyotrophic lateral sclerosis, myopathies

## Abstract

Over the last decade, innate immune system receptors and sensors called inflammasomes have been identified to play key pathological roles in the development and progression of numerous diseases. Among them, the nucleotide-binding oligomerization domain (NOD-), leucine-rich repeat (LRR-) and pyrin domain-containing protein 3 (NLRP3) inflammasome is probably the best characterized. To date, NLRP3 has been extensively studied in the heart, where its effects and actions have been broadly documented in numerous cardiovascular diseases. However, little is still known about NLRP3 implications in muscle disorders affecting non-cardiac muscles. In this review, we summarize and present the current knowledge regarding the function of NLRP3 in diseased skeletal muscle, and discuss the potential therapeutic options targeting the NLRP3 inflammasome in muscle disorders.

## 1. The NLRP3 Inflammasome

### 1.1. NLRP3 Inflammasome Actors

Sterile inflammation drives the pathogenesis of various diseases and is controlled by intracellular multiprotein inflammasome complexes. The latter were described, a decade ago, as a large intracellular signaling platform that contains a cytosolic pattern recognition receptor, such as the nucleotide binding domain leucine rich repeat-containing receptor (NLR) [1]. Among NLR, several members, including NLRP1, NLRP2, NLRP3, NLRC4, NLRP6, NLRP7, and NLRP12, are able to form multimeric inflammasome complexes [2]. Out of them, NLRP3, also known as NALP3, is the best characterized. NLRP3 is a protein coded by the *Nlrp3* (CIAS1) gene and is composed of an amino-terminal pyrin domain (PYD), a central nucleotide-binding domain (NACHT), and a C-terminal leucine rich repeat (LRR) motif [3].

In order to be activated, the NLRP3 inflammasome requires a complex association of specific proteins (Figure 1). The first interaction occurs between NLRP3 and the apoptosis associated speck-like protein containing a C-terminal caspase recruitment domain (ASC). ASC is a cytosolic protein composed of a C-terminal caspase recruitment domain (CARD) and a PYD. This protein plays a central role in the inflammasome complex activation by acting as a bridge between NLRP3 proteins and pro-caspase-1 [4,5]. The interaction of NEK7 with the LRR domain of NLRP3 is also needed to complete inflammasome activation [6]. NEK7 is a member of the NIMA (“never in mitosis gene a”) related serine-threonine kinase family, composed of a catalytic domain with a 30–40 amino acid N-terminal extension [7]. Importantly, in its absence, interleukin-1β (IL-1β) release and the conversion of pro-caspase-1 into its active form, caspase-1, are suppressed [6].

Caspases are a family of proteolytic enzymes that manage the degradation of cellular components during programmed cell death. They are divided into three categories: apoptosis related caspases 2, 3, 6, 7, 8, 9 and 10; inflammation related caspases 1, 4, 5, 11, and 12; and a third category composed of caspases of an unknown function, 13, 14 and 16 [8,9]. Once activated, caspase-1 will, in turn, cleave pro-interleukin-1β (pro-IL-1β), pro-interleukin-18 (pro-IL-18) and Gasdermin D (GSDMD) into their active mature forms [4,10].

IL-1β is a key proinflammatory cytokine involved in the mediation of inflammation in almost every cell type and tissue; where its levels and activities are correlated with the pathogenesis of various autoinflammatory and autoimmune diseases. Interestingly, IL-1β is also able to suppress inflammation and control the adaptive immune responses, Ref. [11] suggesting that the primary physiological function of NLRP3 is to clear noxious substances, and to regulate metabolism and inflammation [12].

IL-18 belongs to the same family of IL-1 cytokines. IL-18 and IL-1β share similar mechanisms of activation, receptor structure, and signal transduction pathways [13]. IL-18 is a pleiotropic cytokine that provides an important link between the innate and adaptive immune responses, and is involved in the regulation of both [14]. Depending on the host microenvironment, IL-18 can be a potent activator of CD4 T helper (Th) 1 lymphocytes and natural killer cells, but also can modulate Th2 and Th17 cell responses, as well as the activity of CD8 cytotoxic cells and neutrophils [14]. Interestingly, IL-1 family members are not secreted via exocytosis as they do not contain a specific secretion signal peptide. As a result, pore creation via GSDMD activation is needed for IL-1 release [15].

GSDMD is a protein composed of a 31 kDa N-terminal (GSDMD-N) and a 22 kDa C-terminal (GSDMD-C) domain, which play an important role in the pyroptosis process. Once recruited by NLRP3 inflammasomes, GSDMD is submitted to a proteolytic cleavage by caspase-1 (canonical) or by caspases 4, 5 and 11 (noncanonical) inflammasome activation [16]. The resulting N-terminal fragment is then released and will oligomerize with other GSDMD-N, thereby creating a complex able to form nonselective pores in the plasma membrane, eventually leading to pyroptosis, as well as to IL-1β and IL-18 release [10,17,18,19].

Pyroptosis is a necrotic cell death type, mediated by GSDMD. In contrast to other types of cell death, such as necrosis and apoptosis, pyroptosis causes the rupture of the cell membrane via pores formation and induces the release of mature inflammatory cytokines, damage associated molecular patterns (DAMPs) and NLRP3 inflammasome specks into the extracellular compartment where they remain active, ultimately resulting in an amplification of the inflammatory response [20,21,22]. Pyroptosis may be a physiologic process during the acute inflammatory response by maintaining cell homeostasis and preventing excessive cell proliferation, but under chronic disease conditions, it will lead to uncontrolled inflammation, excessive cell death and tissue remodeling [23,24].

### 1.2. NLRP3 Activation

Activation of the NLRP3 inflammasome is regulated at both transcriptional and posttranslational levels. First, NLRP3 activation is generated by a priming signal induced by tumor necrosis factor alpha (TNFα), pathogen associated molecular patterns (PAMPs), or IL-1β [25]. All of them act through the nuclear factor (NF)-κB pathway, which, in turn, will upregulate *pro-IL-1β*, *pro-IL 18* and *NLRP3 gene* expression inside the cells, as their levels are otherwise relatively low in numerous cell types [26,27,28]. Besides the innate immunity effectors such as myeloid cells (monocytes and macrophages), muscle cells also contain NLRP3 inflammasomes and thus actively participate in the immune response and eventually to myofiber damage [29,30]. These priming signals also induce posttranslational modifications such as NLRP3 deubiquitination as well as ASC ubiquitination and phosphorylation, allowing inflammasome complex assembly [31].

Subsequently, a second signal is transduced by various activators, such as PAMPs and DAMPs [32]. The latter include viruses [33], adenosine triphosphate (ATP) [34], glycosaminoglycan hyaluronan composing the extracellular matrix [35], amyloid-beta fibrils [36], crystalline structures (including silica, asbestos, aluminum salt, uric acid and calcium pyrophosphate dehydrate (CPPD)) [37], ultraviolet irradiation [38], albumin [39], dietary saturated fatty acids [40], a large number of pore-forming toxins, [41] and aggregates [32,42]. Several molecular mechanisms have been suggested to lead to NLRP3 inflammasome oligomerization. These include the activation of ATP-dependent P2X purino-receptor 7 receptor (P2 × 7R), which causes both Ca^2+^ and Na^+^ influxes, thus leading to increased K^+^ efflux via the opening of the two-pore potassium channel 2 (TWIK-2) [43,44]. In addition, the pore formation induced by bacterial toxins also induces increased K^+^ efflux [43,44,45,46,47,48]. 

Lysosomal alterations [49,50] and mitochondrial reactive oxygen species (ROS) generation [51,52] also promote the complex assembly. More specifically, ROS trigger the activation of NLRP3 in different ways. Firstly, ROS induce a dissociation of thioredoxin interacting protein (TXNIP) and thioredoxin (TRX), which leads to an interaction between TXNIP and NLRP3 [53]. Secondly, ROS cause the release of mitochondrial DNA into the cytosol, ultimately causing NLRP3 inflammasome activation. In addition, ROS lead to NF-κB activity, thereby increasing NLRP3 priming and activation [54].

To date, two pyroptotic pathways are known: the canonical (caspase-1-mediated) and noncanonical (caspases 4, 5 and 11-mediated) inflammasome pathways [55].

The canonical NLRP3 inflammasome pathway is the creation of a multiprotein complex consisting of the previously upregulated NLRP3 protein, an ASC containing a C-terminal caspase recruitment domain, NEK7 and pro-caspase-1. Briefly, the LRR domain of NLRP3 will sense the danger signal, inducing NLRP3 monomers oligomerization. Next, the NLRP3 protein interacts with the PYD of the ASC through homophilic interactions [1]. Afterwards, NEK7 has been demonstrated to bind to the LRR domain of NLRP3 and seems to be necessary for the activation’s completion [6]. Finally, via CARD, the ASC recruits the cysteine protease pro-caspase-1. This association results in the activation of pro-caspase-1 into its cleaved form [56,57,58], leading to the release of mature inflammatory cytokines [59]. In turn, IL-1β and IL-18 will increase the recruitment of white blood cells by activating chemokines and adhesion molecules, leading to leukocyte extravasation. Subsequently, white blood cells also secrete cytokines that induce a sustained inflammatory response, leading to chronic inflammation and increased muscle injury [60].

In addition, the noncanonical activation of the NLRP3 inflammasome, which works in a Toll-like receptor (TLR)/NF-κB pathway independent manner, is achieved via caspases 4, 5, and 11. The latter are activated after cytosolic detection of lipopolysaccharides (LPS), which will open a pannexin-1 (panx-1) transmembrane channel, leading to the release of ATP that open the ATP-gated cation channel receptor (P2 × 7R), thus inducing K^+^ outflow. The ion balance inside and outside the cell membrane is disrupted, implying membrane rupture and intracellular inflammatory content release. These signals then activate the NLRP3 inflammasome [61,62]. Caspases 4, 5 and 11 also specifically cleave the GSDMD into GSDMD-N and GSDMD-C, directly engaging the pyroptosis process. This process will then indirectly activate the NLRP3 inflammasome, thereby increasing the release of inflammatory cytokines and exacerbating the inflammatory response [16]. This noncanonical pathway might lead to endotoxin induced sepsis [61], and is suggested to have a crucial role as a transcriptional factor in promoting Th2 cells differentiation [63] (Figure 2).

### 1.3. NLRP3 Regulation

As previously explained, the primary role of the NLRP3 inflammasome is to modulate inflammation, and its anti-inflammatory effect has been proven in several diseases [64,65]. However, excessive NLRP3 inflammasome activation could also lead to certain autoimmune and metabolic diseases [66]. Therefore, the activation of this inflammasome should be tightly regulated to prevent cell damage and excessive inflammation.

Thus, the regulation of NLRP3 is carried out by many post-translational modifications, such as ubiquitination, phosphorylation, sumoylation and s-nitrosylation [47]. Different parts of the NLRP3 protein can be the target of post-translational modifications creating specific, yet distinct profiles of NLRP3 that can be activated or inhibited. Moreover, the impact of post-translational modifications of NLRP3 is also tissue specific [67].

In addition to post-translational modifications, NLRP3 is also regulated through interactions with partners, called regulators. Among these regulators, many can promote NLRP3 activation, such as TXNIP [53], guanylate binding protein 5 (GBP5) [68], double stranded RNA-dependent protein kinase (PKR) [69,70], migration inhibitory factor (MIF) [71], microtubule affinity regulating kinase 4 (MARK4) [72], as well as Hsp90 and its cochaperon SGT1, the latter also protecting NLRP3 from degradation [73,74]. In contrast, some regulators are involved in the inhibition of NLRP3: they include pyrin- and CARD-only proteins, such as pyrin-only protein 1 (POP1) [75], pyrin-only protein 2 (POP2) [76,77] and inhibitory CARD (INCA also named CARD17) [78].

## 2. NLRP3 and Skeletal Muscle

### 2.1. Introduction

Recently, increasing evidence has highlighted skeletal muscle as having an active and pivotal role in the immune response [30,79,80]. The main evidence leading to this assertion was the ability of muscle cells to secrete their own cytokines, known as myokines. These myokines consist of several hundred secreted proteins and peptides, which may act locally or systemically to mediate numerous metabolic and immune responses [81,82]. Moreover, the detection of the mRNA expression levels of all innate immune receptors in human skeletal muscle biopsies, isolated muscle fibers and primary myotubes confirmed their active involvement as immune effectors [83]. We, too, have provided first evidence for the presence of formed and active NLRP3 within skeletal muscle fibers. NLRP3 inflammasomes were indeed detected as stained clusters in the sarcoplasm of myofibers from wild type (WT) mice challenged by LPS, but not from *Nlrp3* knockout mice [80].

In normal conditions, once activated, the NLRP3/caspase-1/IL-1β pathway activation is a defense mechanism leading to an antiviral, antibacterial, antifungal and anti-inflammatory response that is programmed to cure the pathological tissue [84,85,86,87]. Indeed, in virus mediated disease models, *Nlrp3*-KO mice developed more severe disease than infected WT animals [86]. NLRP3 activation has also been shown to induce leukocyte aggregation and efficient inflammatory responses in Aspergillus fumigatus infected mice [87].

On the other hand, excessive NLRP3 inflammasome activation has been tightly associated with several disorders involving skeletal muscle alterations.

### 2.2. NLRP3 and Skeletal Muscle Diseases

This section is dedicated to muscle diseases where NLRP3 has been described to play a key pathogenic role (Figure 3).

#### 2.2.1. Metabolic Disorders

Metabolic syndrome (MS) encompasses several disorders involving obesity, insulin resistance and type 2 diabetes, hypertension and dyslipidemia leading to cardiovascular disease. Other associated burdens result from ectopic lipid deposits giving rise in liver to non-alcoholic fatty liver disease progressing into non-alcoholic steatohepatitis (NAFLD/NASH), and, in skeletal muscle, to myosteatosis. Hyperuricemia is also often present in this syndrome [88]. All of these components are characterized by chronic low grade inflammation potentially mediated by inflammasomes [89,90,91,92]. The NLRP3 inflammasome may, therefore, be a link between metabolism and inflammation, strengthening the recent concept of metainflammation.

NLRP3 may be a sensor for metabolic “danger” signals, which may either serve as priming signals that induce NLRP3 and pro-IL-1β transcription or as second phase signals triggering inflammasome formation [91,92] (Figure 4).

Priming signals act through the innate immune receptor (toll-like receptor 4), whose ligands include LPS and saturated fatty acids (SFA) [93], or through inflammatory cytokine receptors. Levels of SFA are often increased in MS due to the alleviation of the antilipolytic action of insulin [94], as well as circulating LPS due to dysregulated microbial colonization and ensuing increased gut permeability with the translocation of endotoxins [95]. Priming signals lead to enhanced NF-κB signaling. Like priming signals, second phase signals are also elevated in MS: these include stress molecules such as ATP, ROS and ceramides that activate NLRP3 formation [89,90,91]. Ceramides constitute a subtype of sphingolipids, which are increased in plasma, adipose tissue, liver and the skeletal muscle of animal models with insulin resistance and patients with MS. Ceramide levels negatively correlate with insulin sensitivity [96]; some ceramides may derive from SFAs [91]. Eventually, uric acid can form crystals that are well known NLRP3 activators [90]. By contrast, some molecules may physiologically taper inflammasome responses: adiponectin [82], mono or poly unsaturated fat (n3) [91] or ketone bodies (β-hydroxybutyrate, β-OHB) [97] (Figure 4).

The adipocyte hormone, adiponectin, could simultaneously thwart several facets of the metabolic syndrome by its insulin-sensitizing, fat-burning and anti-inflammatory/antioxidative properties. Adiponectin binds to its receptors: AdipoR1, mainly expressed in skeletal muscle, or AdipoR2, mainly expressed in liver, to activate AMPK or peroxisome proliferator activated receptor (PPAR)-α signaling, respectively, thereby inducing its biological responses. However, adiponectin levels are decreased in obesity and in patients meeting the criteria for the metabolic syndrome [82]. A balanced diet should contain mono or poly unsaturated fat (n3), which may inhibit inflammasome through AMPK or PPAR-γ. Unlike caloric excess, energy deficit, such as starvation, generates metabolic signals, such as β-hydroxybutyrate, that may dampen innate the immune response, which results in sparing energy for the major ketone dependent organs, such as the brain and heart [97] (Figure 4).

Most of the first and second phase signals that enhance NLRP3 responses may either activate the transcription of inflammatory genes via NF-κB or inhibit key components of the insulin-signaling cascade through inactivating phosphorylation, thereby promoting insulin resistance [89,98]. In obesity, adipocyte hypertrophy induces cellular stress together with immune cell infiltration, the release of inflammatory adipokines and insulin resistance. Once adipose tissue storage capacity is overwhelmed, ectopic lipid deposit occurs in several tissues, including the liver and skeletal muscle, further worsening inflammation and insulin resistance [94]. This leads to NAFLD/NASH in the liver and to myosteatosis in skeletal muscle. NLRP3 plays a crucial role in metainflammation and in the interplay between these three insulin target tissues (Figure 5).

Accordingly, *Nlrp3*-KO mice fed a high fat diet (HFD) were protected from adipose tissue inflammation and insulin resistance. Likewise, after weight loss, obese patients showed decreased adipose tissue expression of NLRP3 and IL-1β that was associated with the improvement of insulin sensitivity [90]. Similarly, specific NLRP3 inhibition with MCC950 improved inflammation and insulin sensitivity in a model of mice characterized by type 2 diabetes and dementia [99]. In addition, NLRP3 inhibition by carbenoxolone, a derivative of glycyrrhizic acid, the active ingredient of licorice, markedly reduced intracellular lipid accumulation, inflammation and insulin resistance in liver and skeletal muscle of mice under HFD [100]. The effects on skeletal muscle were further strengthened by in vitro experiments. Lipids within muscle fibers mainly stored in neutral lipid droplets (LDs) are coated by lipid droplet associated proteins, which are referred to as perilipins (PLIN). One of these, PLIN2, is a marker for LDs in human skeletal muscle, and the levels of intramuscular PLIN2 and triglycerides are closely correlated. PLIN2 overexpression in C2C12 myotubes was accompanied by the activation of the NLRP3 inflammasome, which led to impaired insulin induced glucose uptake, while the siRNA gene silencing of NLRP3 remedied this effect [101]. In humans, a simple change from the habitually high palmitic acid (SFA) intake into a high oleic acid diet, oleic acid being an omega-9 monounsaturated fatty acid (MUFA), resulted in a lower secretion of IL-1β, IL-18, and TNFα, as well as less NLRP3 mRNAs in skeletal muscle, potentially lowering the prevalence of insulin resistance and type 2 diabetes [102].

#### 2.2.2. Muscle in Aging/Sarcopenia

As society ages, the incidence of physical limitations is dramatically increasing, thereby enhancing the risk of falls, institutionalization, comorbidity, and premature death. An important cause of physical limitations is the age related loss of skeletal muscle mass and function, also referred to as sarcopenia [103,104]. Beyond physical performance, muscles also play a crucial role in insulin sensitivity and fuel homeostasis. Muscle disturbances may, thus, lead to insulin resistance and metabolic disorders [105].

Overall loss of skeletal mass results from an imbalance between muscle protein anabolic and catabolic pathways, where protein synthesis is hindered and protein breakdown is excessive [106]. However, the cellular and molecular mechanisms underlying sarcopenia are still poorly understood, although this condition is currently considered to be multifactorial [107]. The aging process is associated with a decline in autophagic capacity which impairs cellular housekeeping, leading to protein aggregation and the accumulation of dysfunctional mitochondria, which provoke ROS production and oxidative stress. These danger signals, in turn, activate inflammasomes which provoke a low grade inflammation in several tissues, referred to as inflammaging. This further inhibits autophagy and accelerates the aging process [108,109]. Chronic inflammation together with reduced muscle mass could also promote insulin resistance. Insulin resistance, inflammatory cytokines, the inhibition of autophagic capacity, mitochondrial dysfunction and ROS induce and perpetuate inflammaging and lead to sarcopenia [110]. In addition, inflammaging could aggravate several other age related degenerative changes [108,111].

The involvement of inflammasomes in ageing and age related diseases [112], including sarcopenia, has been strengthened by studies in mice with genetic deletion of Nlrp3. Thus, deletion of the NLRP3 inflammasome enhances healthspan and protects against insulin resistance, bone loss, reduced cognitive function and motor performance [113]. Ageing is also associated with decreased skeletal muscle strength and slowing of movement, in which increased NLRP3-dependent caspase-1 activity in muscle is described. The deletion of mouse Nlrp3 prevented the reduction in muscle mass, increased muscle strength and endurance and protected from age related increases in the number of myopathic fibers [114]. Another study confirmed these data and further showed a reduction in fibrosis and apoptotic nuclei in the skeletal muscles of aged *Nlrp3*-KO mice, compared to wild-type ones, as well as less mitochondrial damage and multivesicular bodies resulting from defective autophagy/mitophagy [115]. Oral administration of melatonin, a pineal hormone with antioxidant and anti-inflammatory properties, in a mouse model of sarcopenia was also shown to reduce the expression of pro-caspase 1 mRNA and to preserve the normal muscular structure and activity of skeletal muscles [116,117,118].

#### 2.2.3. Critical Limb Ischemia

Critical limb ischemia (CLI) is the most severe clinical manifestation of peripheral arterial disease, where failure to establish revascularization eventually leads to amputation and even to death [119]. Metabolic syndrome and ageing, both characterized by low grade inflammation, are risk factors for the development of peripheral arterial disease and are known to impair skeletal muscle postischemic vascular recovery [120]. However, the molecular mechanisms involved are still unknown. A hypothesis suggests that NLRP3 could also be implicated in mediating inflammation and angiogenesis after ischemia. In a hind limb ischemia mouse model, blood flow and vascular density were impaired in HFD mice. This was carried out through TXNIP-dependent NLRP3 inflammasome activation in muscle, which led to significant increases in active caspase-1 and IL-1β and compromised vascular recovery in response to ischemia. Targeting the NLRP3 inflammasome by using *Txinpt*-KO mice mitigated HFD-induced inflammation and impaired angiogenesis, thus opening a potential therapeutic target in obesity induced vascular complications [121].

Ischemic murine muscle also exhibited a reduced expression of a specific circular RNA (circHIPK3). Treatment with exosomes delivering circHIPK3 into skeletal muscle reduced ischemia induced pyroptosis caused by inflammasome, as evidenced by less activation of NLRP3, cleaved caspase-1, and reduced increase in IL-1β and IL-18. Accordingly, this treatment improved blood perfusion, running distance and muscle force in mice. Taken together, these data indicate that inhibition of inflammasome and pyroptosis prevents hindlimb ischemic injury [122].

Finally, heme oxygenase-1 (HO-1) could also be a critical player in inducing NLRP3 in ischemic muscle. While postischemic inflammation is needed for initiation of neovascularization, excessive inflammatory response suppresses perfusion recovery. HO-1 is an immunomodulatory enzyme primarily expressed in macrophages [123]. A recent study found the upregulation of HO-1 expression in murine muscle after hindlimb ischemia surgery which mainly occurred in infiltrated macrophages. Suppressing HO-1 was able to restore blood flow, motor function and attenuate tissue damage in muscles after hindlimb ischemia. This was carried out by reducing NLRP3 inflammasome activation and accelerating its autolysosomal degradation. Moreover, inhibiting inflammasome activation with i.p. MCC950 improved blood flow and capillary density in mice, therefore underlining the importance of NLRP3 in ischemic muscle diseases [124].

#### 2.2.4. Sepsis Induced Muscle Atrophy

Sepsis is an excessive response of the body against an infection leading to tissue and organ damage. This condition requires intensive care management and accelerates muscle atrophy in bed bound patients [125]. Muscle wasting is linked to the inflammatory response occurring during the acute phase of sepsis, and might potentially be mediated by the NLRP3 inflammasome [126].

The skeletal muscles of mice subjected to acute inflammation by intraperitoneal LPS injection displayed significantly high levels of inflammatory components, such as the NLRP3 inflammasome and IL-1β. This was accompanied by an increase in muscle atrophy signaling pathways. The Forkhead box O (FoxO) family of transcription factors plays a critical role in protein breakdown by activating the expression of atrogenes (which include two muscle specific ubiquitin ligases, atrogin-1 and MuRF1) responsible for profound loss of muscle mass. All of these components were upregulated in the muscles of LPS mice. These deleterious effects were abolished in mice pretreated with an inhibitor of a double strand RNA-dependent protein kinase (PKR), which blocks inflammatory cytokine expression [127]. Likewise, Triptolide, a plant derivative previously described as an NLRP3 inhibitor [128], attenuated LPS-induced myotube atrophy in vitro in C2C12 cells and in vivo in LPS injected mice. Thus, triptolide decreased plasma inflammation while increasing skeletal muscle weight, strength and locomotion, thereby preventing muscle atrophy in LPS challenged mice [129]. OLT1177, an orally active β-sulfonyl nitrile molecule targeting the NLRP3 NACHT domain, was also shown to reduce the severity of systemic inflammation in mice challenged with LPS, where muscle IL-1β and oxidative stress were lowered [130]. Moreover, *Nlrp3*-KO mice, submitted to polymicrobial sepsis induced by cecal ligation and puncture surgery, had a survival benefit and did not lose body or muscle weight during 96 h of sepsis, when compared to wild type ones. This was associated with a reduction in IL-1β serum levels [131]. In humans, administration of the ketone body β-hydroxybutyrate (β-OHB) reduced muscle protein breakdown after LPS injection, indicating that β-OHB exerts anticatabolic effects during acute inflammation [132]. Taken together, all these data reinforce the idea that NLRP3 plays a crucial role in inflammation induced muscle atrophy in sepsis.

Finally, blocking gasdermin D pore formation by Disulfiram treatment (an approved drug used to treat alcohol addiction, see last chapter) tapered LPS induced sepsis in mice: circulating levels of inflammatory cytokines were reduced and survival was greatly improved. One advantage of this drug is to block LPS induced inflammasome activation by both noncanonical and canonical pathways [133].

#### 2.2.5. Inherited Myopathies

Inherited myopathies are a heterogeneous group of diseases primarily affecting the skeletal muscle tissue. These are caused by mutations in different genes encoding proteins that are critical for muscle structure and function [134] (https://rarediseases.info.nih.gov/, accessed on 12 October 2021). They are characterized by progressive muscle weakness and wasting, along with a severe and persistent muscle inflammation that plays a central role in the onset and progression of these diseases [82]. Various studies have demonstrated that the NLRP3 inflammasome triggers a pathogenic inflammatory response in many inherited myopathies, including limb girdle muscular dystrophy type 2B (LGMD2B) [30], valosin-containing protein (VCP) associated diseases [79], and Duchene muscular dystrophy (DMD) [80,135].

LGMD2B is one type of limb-girdle muscular dystrophy, a group of heterogeneous diseases that affect the voluntary muscles. LGMD2B is caused by mutations in the dysferlin gene, which encodes a protein that is thought to aid in repairing the muscle fiber membrane when it becomes damaged or torn. LGMD2B is a slowly progressive disease that causes muscle weakness and atrophy, mainly of the pelvic muscles and muscles of the shoulder girdle [136] (https://rarediseases.info.nih.gov/, accessed on 12 October 2021). Rawat and colleagues were the first to show that, besides immune cells, primary skeletal muscle cells expressed TLRs and can efficiently produce and secrete IL-1β in stressful conditions. They also showed that inflammasome components were significantly up regulated in dysferlin deficient muscle cells, and likely contributed to the pathogenesis of LGMD2B [30].

VCP is a newly identified calcium associated ATPase protein that has been associated with various degenerative disorders that encompass inclusion body myopathy, Paget’s disease of bone, and frontotemporal dementia. VCP disease is a rare and progressive neuromuscular disorder, with death typically occurring in the 50s and 60s from respiratory and cardiac failure [137] (https://rarediseases.info.nih.gov/, accessed on 12 October 2021). Recently, the NLRP3 inflammasome and IL-1β were found to contribute to the pathogenesis of VCP associated myopathies. Indeed, NLRP3 was active in primary cultures of myoblasts derived from VCP patients, in quadriceps muscles of VCP mice, and in the inflammatory macrophages that infiltrated those muscles. Treatment with MCC950 reversed NLRP3 activation, both in vitro and in vivo, and significantly ameliorated the muscle strength of VCP mice [79].

DMD is the most frequently inherited human myopathy and the most devastating type of muscular dystrophy. DMD is caused by mutations in the gene encoding for dystrophin, a key scaffolding protein, which forms an important protein complex that connects the actin cytoskeleton of myofibers to the extracellular matrix (Figure 5). This complex is crucial for maintaining cell membrane stability and permeability, as well as normal contractile function of the skeletal muscle. Absence of dystrophin leads to the disruption of this complex and, thus, to membrane damage, allowing for DAMP release, chronic inflammation and severe muscle degeneration. DMD remains a lethal muscle disorder with no cure, where the first signs of muscle weakness begin early on in life and, without proper intervention, death typically occurring in the 20s and 30s [82,138]. Our group has shown that NLRP3 and IL-1β were highly expressed, not only in C2C12 murine cell lines challenged by inflammation but, also, in human primary cultured myotubes derived from DMD patients. Likewise, all inflammasome components were upregulated in the skeletal muscles of mdx mice (a mouse model of Duchenne), and the complex was overactivated [80]. Specific NLRP3 depletion in mdx mice markedly protected the skeletal muscle against inflammation, oxidative stress and injury, while increasing its force and endurance, thus helping to rescue the dystrophic phenotype [80,82]. Moreover, we discovered that adiponectin, a pleiotropic adipokine with potent anti-inflammatory effects, could significantly downregulate the NLRP3 inflammasome [80]. Briefly, adiponectin binds to its muscle specific receptor, AdipoR1, and activates the AMPK pathway, which, in turn, represses the NLRP3 inflammasome, in part through reduction of NF-κB activity and oxidative stress [82,138]. In addition, activating AMPK leads to an upregulation of the adiponectin muscle anti-inflammatory mediator, miR-711. We found that adiponectin, through miR-711, could be a major repressor of the NLRP3 inflammasome by inhibiting both its priming and activation in muscle [80,82]. This repression occurred both in vitro, in C2C12 myotubes, and in vivo after either local or systemic adiponectin supplementation [80,139] (Figure 6). In agreement with our data, other drugs and molecules that mainly activate AMPK signaling, such as AICAR (analog of adenosine monophosphate), resveratrol, a natural polyphenolic compound, and metformin (used for treating type 2 diabetes) were also found to mitigate some features of DMD in cell cultures and in animal models [140,141,142,143]. Metformin was also tested in a randomized, double-blind, placebo-controlled Phase III clinical trial in combination with L-citrulline, as a possible treatment for DMD. The study is completed and it showed only a small reduction in motor function decline among the stable subgroup of patients treated with this combination therapy (ClinicalTrials.gov Identifier: NCT01995032, accessed on 2 November 2021). A big part of their beneficial effects is mediated through the reduction in inflammation and in the inflammasome [112,138]. Similar to adiponectin, the gastric peptide ghrelin, famously known as the “hunger hormone”, is another circulating hormone with an anti-inflammatory effect [144]. Once injected in mdx mice, ghrelin was found to improve muscle performance and alleviate muscle pathology through the inhibition of NLRP3 inflammasome activation and subsequent maturation of IL-1β [135]. In addition, the plant compound, curcumin, a NF-κB inhibitor, also showed beneficial effects on the dystrophic skeletal muscle by reducing the levels of TNFα and Il-1β and improving cell membrane integrity, once injected in mdx mice [145] (Figure 6).

Finally, we are currently investigating a possible beneficial and therapeutic effect of MCC950, a specific NLRP3 inhibitor, on the pathogenesis of DMD using both in vivo mdx mice and in vitro primary cultures of human DMD myotubes. Preliminary results look promising and show an improvement in muscle performance and protection against muscle inflammation [146] (Figure 6).

#### 2.2.6. Acquired Myopathies

Idiopathic inflammatory myopathy (IIM) is an acquired immune mediated muscle disease including dermatomyositis (DM), polymyositis (PM), sporadic inclusion body myositis (sIBM) and juvenile dermatomyositis (JDM) [147]. IIM is characterized by myalgia, muscle weakness, and extramuscular manifestations. Muscle biopsies show alterations such as necrosis, atrophy and sometimes inflammatory infiltrates [148,149,150].

To date, the pathogenesis of IIM remains unclear. However, several pieces of evidence indicate that the NLRP3 inflammasome may be involved in muscle damage. Recent studies have shown the direct impact of the canonical and noncanonical pathways of pyroptosis in the occurrence and progression of IIM [60,151]. In the experimental autoimmune myositis (EAM) mice model, serum levels of IL-1β and IL-18, as well as mRNA expression and the protein levels of NLRP3, GSDMD, Caspase 11 and P2X7R were increased [60]. The implication of the NLRP3 inflammasome has also been confirmed in humans, where IL-1β and IL-18 were shown to be highly expressed in the muscle and serum of DM and PM patients [29,152,153]. Moreover, DM and PM patients displayed a higher protein expression of NLRP3 and caspase-1 in muscle tissues, in comparison with controls [29,153]. Finally, N-GSDMD, the executioner of pyroptosis, was also upregulated in PM patients [150]. Taken together, these results confirm the potentially pivotal regulatory role of the NLRP3 inflammasome in IIM genesis and progression.

Several molecular pathways have been proposed to explain NLRP3 inflammasome activation in IIM pathogenesis. Firstly, TNFα activates NF-κB signaling leading to the canonical activation of the NLRP3 inflammasome [154]. Secondly, ROS trigger the activation of the NLRP3 inflammasome through TXNIP and the release of mitochondrial DNA [54]. Thirdly, mTOR pathway, via mTORC1 activation, stimulates IL-1β expression and maturation through hypoxia inducible factor-1α (HIF-1α), while rapamycin, a selective inhibitor of mTORC1, reduces NLRP3 and IL-1β levels [60]. Moreover, hypoxia upregulates the high mobility group box 1 protein (HMGB1), leading to the activation of NF-κB pathway and of NLRP3/IL-1β axis, thus triggering an inflammatory response [60].

Finally, glucose metabolism dysregulation could also contribute. Indeed, 18-fluorodeoxyglucose positron emission tomography/computerised tomography (PET/CT) showed abnormal glucose uptake in muscle tissues of patients with IIMs [155]. These observations were then explained by the direct implication of the glycolysis in the muscle damage process in IIM. Indeed, upregulation of pyruvate kinase isozyme M2 (PKM2) was shown in DM and PM compared with controls. Moreover, muscle PKM2 expressions were correlated with NLRP3 inflammasome expression levels [150], confirming a relationship between NLRP3 and dysregulated glucose metabolism [156].

Taken together, these results suggest that NLRP3 is involved in the immune response and myofiber alteration in IIMs.

#### 2.2.7. Amyotrophic Lateral Sclerosis

Amyotrophic lateral sclerosis (ALS) is a neurodegenerative disease characterized by progressive muscle weakness and atrophy, ultimately leading to death within approximately 2 to 5 years. The exact pathogenesis of ALS is still unknown. However, inflammation has been shown to be a prominent pathological finding in ALS patients [157] and a few studies suggest that the NLRP3 inflammasome might have a pivotal role in ALS. Indeed, the activation of the NLRP3 inflammasome has been observed, in brain, spinal cord and in the skeletal muscle of SOD1G93A mice, a transgenic mouse model of ALS expressing a mutant form of human Superoxide Dismutase 1 (SOD1), [158,159], as well as in those of sporadic ALS (sALS) patients [160,161,162]. NLRP3 mRNA levels were also significantly elevated in the white blood cells of sALS patients, compared to healthy controls [12]. In addition, mRNA levels of ASC, caspase 1 and IL-1β were increased at the asymptomatic stage in skeletal muscles of SOD1G93A mice [12,162], whereas their respective protein expression was still normal. However, in the later stage of the disease, increased protein levels of inflammasome components were observed [12]. Therefore, a link between NLRP3 inflammasome activation and ALS disease progression is thought to exist.

This early involvement of muscle revised our preconceived idea, in which muscle alterations are only the consequence of motoneuron destruction. Regarding the current data, skeletal muscle is increasingly considered as an active player in ALS pathogenesis. Indeed, as explained before, skeletal muscle expresses different PRRs, allowing a muscle specific response to environmental factors [83,163]. Moreover, primary muscle cells may release IL-1β after treatment with LPS and ATP confirming its primary role in inflammasome activation [30,80]. Muscle inflammation might then activate a retrograde signaling, leading to motoneuron death [164,165,166].

Interestingly, the NLRP3 inflammasome may play a dual role in ALS pathogenesis. Indeed, at an early stage of the disease, the NLRP3 inflammasome may exert positive effects. Thus, a positive correlation was observed between Nlrp3 mRNA levels in skeletal muscle and lifespan in SOD1G93A mice [12]. This might be partially explained by the fact that NLRP3, independently from its role in inflammasome, can act as a transcription factor in Th2 lymphocytes, promoting the expression of IL-4 [167], a cytokine responsible for muscle growth and regeneration [168]. The positive effect of NLRP3 on mouse lifespan was confirmed, as mice that did not receive MCC950, a selective inhibitor of NLRP3, lived longer [12]. These paradoxical results display the early role of the NLRP3 inflammasome, which is to clear noxious protein aggregates, a characteristic feature of ALS [169].

Taken together, these results suggest the two faced action of the NLRP3 inflammasome in ALS: where, at the early stage of the disease, it plays a beneficial role by clearing noxious aggregates, while, when the disease progresses, the chronic NLRP3 stimulations by an excess of damage signals, such as mutant proteins SOD1 and TDP-43, change its positive effect into a harmful action leading to myofiber damage and, ultimately, to motoneuron degeneration [169,170].

## 3. Therapeutic Perspective Targeting NLRP3

The NLRP3 inflammasome unequivocally plays a key pathological role in the development and progression of several skeletal muscle disorders. As a result, treatments targeting the NLRP3/caspase-1/IL-1β axis are expected to improve our drug arsenal to combat muscle diseases with an excessive and deleterious inflammatory component.

Therapies could potentially act either directly on NLRP3 protein (direct inhibitors) or interact with the upstream or downstream NLRP3 signaling pathways (indirect inhibitors).

### 3.1. NLRP3 Direct Inhibitors

We will review some direct inhibitors typically tested in inflammatory diseases and emphasize the ones already tested on muscle related disorders. These inhibitors are summarized in Table 1.

#### 3.1.1. Inflammatory Disorders

As mentioned, many specific NLRP3 inhibitors have been described in a plethora of inflammatory diseases in the past years, such as Glitazone (Cy-09), 3,4-methylenedioxy-beta-nitrostyrene (MNS), MCC950, Dapansutrile (OLT1177), Tranilast, and Oridonin. All of these inhibitors either target NLRP3 ATPase activity [130,171,172,173,174] and/or specifically block NLRP3 oligomerization [175,176]. INF39 and β-OHB also act directly on the NLRP3 protein, however their exact mechanism of action is still unknown [132,177]. Finally, ZYIL1, a new oral small molecule, has been shown to prevent NLRP3 induced ASC oligomerization. Moreover, a phase I clinical trial in healthy human volunteers is currently ongoing with this molecule (ClinicalTrials.gov Identifier: NCT04972188).

#### 3.1.2. Skeletal Muscle Disorders

Among these direct inhibitors, three molecules have recently displayed promising results in skeletal muscle.

MCC950 is an extremely potent inhibitor of NLRP3, by binding the NLRP3 NACHT domain thereby blocking ATP hydrolysis [173]. MCC950 was shown to rescue neonatal lethality in a mouse model of NLRP3 activating mutation, while the targeted blockade of IL-1β alone was unable to do so. This compound was also active in ex vivo samples from patients with a similar gain of function mutations [178,179]. Moreover, MCC950 was shown to improve blood flow and capillary density in mice, confirming the importance of NLRP3 in ischemic muscle diseases [124]. In addition, in a mouse model of VCP myopathy, MCC950 increased mice physical performances and significantly reduced NLRP3, caspase 1, IL-1β and IL-18 expression [79]. Preliminary results of MCC950 tested on the pathogenesis of DMD look very promising, and also show an improvement in muscle performance and a protection against muscle inflammation [146]. Therefore, MCC950 could present a promising therapeutic option for many muscle disorders.

The above mentioned β-OHB exerts protective muscular anticatabolic effects in volunteers submitted to LPS [132], while OLT1177 reduces systemic and muscle inflammation in LPS challenged mice [130]. A phase II clinical trial is currently ongoing for the treatment of moderate COVID-19 with OLT1177 (ClinicalTrials.gov Identifier: NCT04540120, accessed on 12 October 2021).
cells-10-03023-t001_Table 1Table 1Direct NLRP3 inhibitors, their mechanisms of action and involvement in diseases.AgentTarget SiteInhibitory EffectTestedon SMDiseasesClinical TrialsReferencesCy-09NLRP3NACHTDomainNLRP3 ATPase activity−Gout, T2D, CAPS−[172]MCC950+Multiple sclerosis, CAPS, Ischemia, CLI, VCP, DMD−[60,79,173]MNS−Inflammatorydiseases−[171]OLT1177+Arthritis, CAPS,GoutArthritis, Phase IICovid19, Phase II[130]OridoninNLRP3 oligomerization−T2D, gout-[176]Tranilast−Gout, T2D, CAPSCAPS, Phase II[175]β-OHBUnknown+/*ALSALS, Phase II[97,132]ZYIL1−−Phase INo publicationINF39NLRP3 ATPase activity and oligomerization−Inflammatory bowel disease−[177]β-OHB: β-hydroxybutyrate; ALS: amyotrophic lateral sclerosis; CAPS: cryopyrin associated periodic syndrome; Cy-09: glitazone Cy-09; DMD: Duchenne muscular dystrophy; MNS: 3,4-methylenedioxy-beta-nitrostyrene; NLRP3: NOD like receptor family, pyrin domain containing 3; OLT1177: Dapansutrile; SM: skeletal muscle; T2D: type 2 diabetes; VCP: valosin-containing protein; +: validated; − absence; * Not directly demonstrated.


### 3.2. NLRP3 Indirect Inhibitors

We will focus herein merely on indirect inhibitors tested in skeletal muscle.

#### 3.2.1. NLRP3 Upstream Inhibitors in Skeletal Muscle

Several drugs and molecules have been described as inhibitors of inflammasomes by acting upstream of NLRP3 oligomerization (Table 2).

Among them, two compounds target the P2X7/K^+^ channel activation pathway: bright blue G (BBG) [60,180,181] and Glyburide (a sulfonylurea currently used in T2D) [60,182], both displaying a restoration of muscle strength in IIM mouse models.

NF-κB mediators have shown interesting results in skeletal muscle inflammatory models. As previously explained, triptolide prevented muscle atrophy in LPS challenged mice [129], carbenoxolone was able to decrease metabolic abnormalities, such as liver and muscle steatosis, in HFD-mice [100], while melatonin showed an interesting anti-inflammatory effect and preserved the normal muscular structure and activity in a sarcopenic mouse model [116,117,118,183]. Curcumin was shown to decrease ROS levels and proinflammatory cytokines in C2C12 muscle cells submitted to palmitate induced inflammation [184], and to improve the dystrophic phenotype in mdx mice [145]. Several other molecules, such as adiponectin, AICAR, metformin, and resveratrol, were also found to mitigate some pathological features of DMD in cell cultures and animal models, mainly by activating AMPK signaling thus inhibiting NF-κB and reducing inflammasome activation [138,140,141,142,143].

In addition, ghrelin was shown to inhibit NLRP3 inflammasome activation, reduce muscle pathology and enhance muscle performance [135]. Shikonin, a Chinese medicine inhibitor of pyruvate kinase M2 (PKM2), exhibited NLRP3 inhibitory activation and protected from pyroptosis in muscle cells [150,185]. Finally, human volunteers under a high palmitate diet (SFA) displayed high levels of NLRP3 mRNAs in their skeletal muscle biopsies, while switching to an oleate rich diet (MUFA) reduced NLRP3 priming and activation [102].

Although indirect inhibitors are able to suppress NLRP3 inflammasome activation, high doses are nonetheless required. Thus, these indirect inhibitors seem less sensitive and effective than direct ones. In addition, the mechanisms of action for some of these molecules can be tissue specific, making them less impressive in diseases where multiple organs and tissues are affected. Therefore, directly targeting NLRP3 probably represents the best therapeutic approach for muscle diseases with an inflammatory component.
cells-10-03023-t002_Table 2Table 2List of indirect inhibitors acting upstream of NLRP3 priming in skeletal muscle.AgentMechanism of ActionEffectRelevant Clinical TrialReferencesAdiponectinActivation of AMPK signaling pathwayReduction in NF-κB activity leading to downregulation of NLRP3 and proinflammatory cytokine expression−[80,138]AICAR−[143]ResveratrolCHFC, Phase IIMetabolic Syndrome, Phase II[142]Oleic acid (MUFA)−[102]MetforminActivation of AMPK pathway and inhibition of TLR4 signaling pathwayCommercialised for T2D, Phase III[140]BBGInhibition of P2X7R pathway/K^+^ outflow−[60]GlyburideCommercialised for T2DCarbenoxoloneDecreased phosphorylation ofIκBα−[100]Triptolide−[129]CurcuminDecreased phosphorylation ofIKKα-IKKβT2D, Phase IV[145,184]GhrelinInhibition of JAK2-STAT3 and p38 MAPK signaling pathway−[135]MelatoninInduction of SIRT1 deacetylase activity through RORα-dependent mechanisms−[115,183]ShikoninInhibition of PKM2Downregulation of NLRP3 and proinflammatory cytokine expression by unknown mechanism−[150]AICAR: 5-Aminoimidazole-4-carboxamide ribonucleotide; AMPK: AMP-activated protein kinase; BBG: bright blue G; CHFC: congestive heart failure chronic; IκBα: nuclear factor of kappa light polypeptide gene enhancer in β-cells inhibitor, α; IKK: inhibitory-κB kinase; JAK2: Janus kinase 2; MAPK: mitogen activated protein kinases; MUFA: mono unsaturated fatty acids; NLRP3: NOD like receptor family, pyrin domain containing 3; P2X7R: ATP gated cation channel receptor; PKM2: pyruvate kinase isozyme M2; NF-κB: nuclear factor-κB; SIRT1: sirtuin 1; STAT3: signal transducer and activator of transcription 3; T2D: type 2 diabetes; TLR4: toll-like receptor 4; TNFα: tumor necrosis factor α; − absence.


#### 3.2.2. NLRP3 Downstream Inhibitors in Skeletal Muscle

Other drugs and molecules could act downstream of the NLRP3 inflammasome to inhibit pyroptosis and/or inflammation.

Disulfiram, as previously explained, acts by blocking gasdermin D pore formation and shows a reduction in inflammatory cytokines level in a mouse model of sepsis associated with a longer survival rate. [133].

Anti-IL-1β therapies were the first to be tested in humans and showed efficacy in several inflammatory diseases. However, their effects in metabolic disorders were less impressive [186,187]. Moreover, the use of canakinumab, a human monoclonal antibody targeted at IL-1β, was also associated with increased susceptibility to infection [188].

Anti-IL-18 therapies are currently in development for different inflammatory diseases. Tadekinig Alfa, a recombinant human IL-18 binding protein, is being tested in adult onset Still’s disease and in NLRC4 related macrophage activation syndrome (inflammatory diseases associated with high plasma IL-18 levels) (ClinicalTrials.gov Identifier: NCT02398435, NCT03113760). In addition, GSK1070806, a humanized antibody targeting IL-18, is currently being tested in a phase 1 trial for atopic dermatitis (ClinicalTrials.gov Identifier: NCT04975438). To our knowledge, these medications have not yet been tested on skeletal muscle related disorders.

It is also important to note that IL-1β or IL-18 specific sequestration by a pharmacological approach do not prevent pyroptosis [10,13], thereby limiting their effectiveness to only one side of the inflammasome activation downstream effects. Therefore, the development of new therapeutics directly targeting the NLRP3 inflammasome is much needed.

## 4. Conclusions

In summary, the NLRP3 inflammasome displays a primary protective function in the muscle by clearing noxious substances. However, its excessive activation has been identified to play a key pathological role in the development and progression of several skeletal muscle diseases.

Our continuous understanding of physiological and pathological processes of inflammation is leading to the development of novel therapeutic approaches that target the NLRP3 inflammasome and that are showing promising results for several pathological conditions.

Finally, the existence of NLRP3 inhibitors entering the pipelines of human clinical trials for several inflammatory diseases, will surely pave the way for future trials for muscle related diseases with an inflammatory component.

## Figures and Tables

**Figure 1 cells-10-03023-f001:**
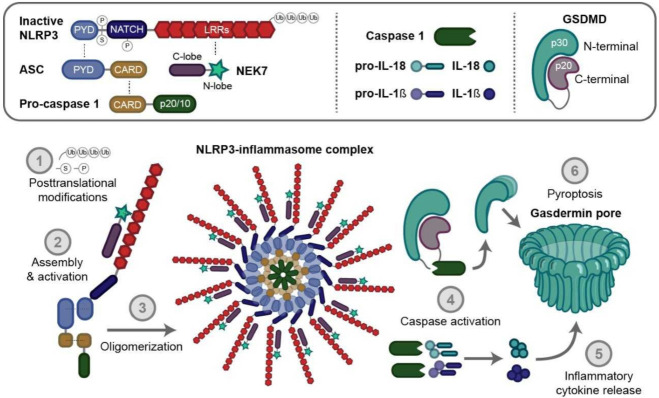
The nucleotide-binding oligomerization domain like receptor family (NOD-like) pyrin domain containing 3 (NLRP3) inflammasome complex. The NLRP3 inflammasome is a multiprotein complex regrouping several proteic actors. In basal conditions, NLRP3 is selfrepressed via an internal interaction between its NACHT and LRR domain. ‘S’, ‘P’, and ‘Ub’ symbols on the inactive form of NLRP3 mean SUMOylation (SUMO, Small Ubiquitin-like Modifier proteins), phosphorylation and ubiquitination, respectively. After its priming, several posttranslational modifications take place (1), leading to the removal of this interaction and the binding of NLRP3 to ASC protein, through its PYD domain, and to NEK7 via its LRR domain, and, thus, NLRP3 activation. In turn, procaspase-1 interacts with ASC through its CARD domain (2). Oligomerization (3) triggers the activation of caspase-1 (4) and the processing of pro-IL-1β, pro-IL-18 and Gasdermin D (GSDMD) into their mature forms, thereby triggering inflammatory cytokines release (5) and pyroptosis (6). ASC, apoptosis associated speck like protein containing a caspase recruitment domain; CARD, caspase recruitment domain; LRR, leucine-rich repeats; NACHT, NAIP (neuronal apoptosis inhibitor protein), C2TA (MHC class 2 transcription activator), HET-E (incompatibility locus protein from *Podospora anserina*) and TP1 (telomerase-associated protein) NLRP3, NOD-, LRR- and pyrin domain-containing protein 3; PYD, pyrin domain.

**Figure 2 cells-10-03023-f002:**
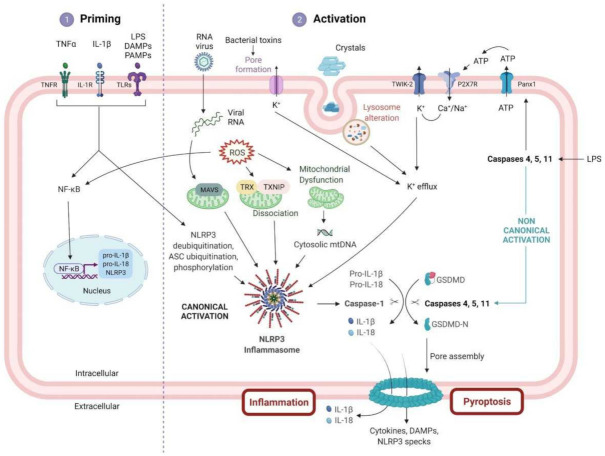
NLRP3 inflammasome priming and activation pathway. Activation of the NLRP3 inflammasome signaling pathway requires two signals. The signal ① or priming (**left**) is triggered by the stimulation of cytokine receptors (TNFR and IL-1R) or TLRs via several activators, such as LPS, DAMPs or PAMPs. This interaction with their respective receptors will then lead to the upregulation of several inflammasome components (NLRP3, pro-IL-1β and pro-IL-18) via the NF-κB pathway as well as to posttranslational modifications of ASC and NLRP3 protein. Signal ② or activation (**right**) is triggered by numerous PAMPs or DAMPs including virus, crystals, protein aggregates, LPS, bacterial toxins, or ATP, which will, in turn, activate multiple signaling cascades. The latter involves, among others, K^+^ efflux, ROS production, lysosomal dysfunction, and the release of mitochondrial DNA in the cytosol. These signals lead eventually to the interaction between NLRP3, ASC and pro-caspase-1, allowing the formation of the NLRP3 multiprotein inflammasome complex. Once activated, caspase 1, the head of the canonical activation, cleaves the pro-IL-1β, pro-IL-18 and Gasdermin D (GSDMD) into their mature forms, triggering pyroptosis and inflammatory cytokines release. In addition, another way leading to pyroptosis is the noncanonical pathway, where cytosolic lipopolysaccharide (LPS), via caspases 4, 5 and 11, can also cleave GSDMD and form pores into the cellular membrane. IL-1R1, IL-1 receptor type 1; PANX1, pannexin 1; ROS, reactive oxygen species; TNFR, tumor necrosis factor receptor; TRX, thioredoxin.

**Figure 3 cells-10-03023-f003:**
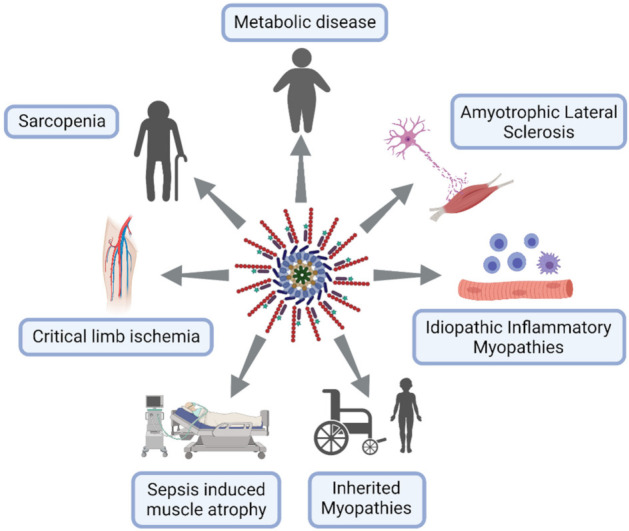
NLRP3 inflammasome excessive activation in skeletal muscle diseases. Lymphocytes in blue, macrophages in purple and neuron in pink.

**Figure 4 cells-10-03023-f004:**
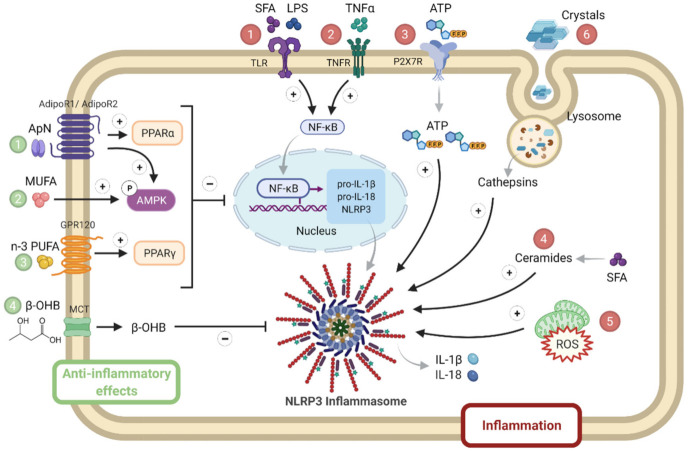
NLRP3 priming and activation in metabolic syndrome. Several activators of either priming or NLRP3 complex formation are numbered in red. They include SFA, LPS ①, inflammatory cytokines such as TNFα ②, stress molecules such as ATP ③, ceramides ④ and ROS ⑤, as well as crystals ⑥. By contrast, inhibitory hormonal or metabolic signals are numbered in green: they include adiponectin (1), MUFA (2) or n-3 PUFA (3), or ketone bodies (β-OHB) (4). Adiponectin binds to AdipoR1 (predominantly expressed in skeletal muscle) or to AdipoR2 (predominantly expressed in liver) to activate AMPK or PPAR-α signaling, respectively. MUFA and PUFA activate AMPK and PPAR-γ signaling, respectively. β-OHB is produced by the liver and is used as fuel by other tissues (such as muscle), where it enters via a MCT. β-OHB, β-hydroxybutyrate; MCT, monocarboxylate transporter; MUFA, mono unsaturated fatty acids; PPAR, peroxisome proliferator activated receptor; PUFA, poly unsaturated fat; SFA, saturated fatty acids. Symbols in circles indicate: ‘P’, phosphorylation; ‘+’, activation; and ‘−’, inhibition.

**Figure 5 cells-10-03023-f005:**
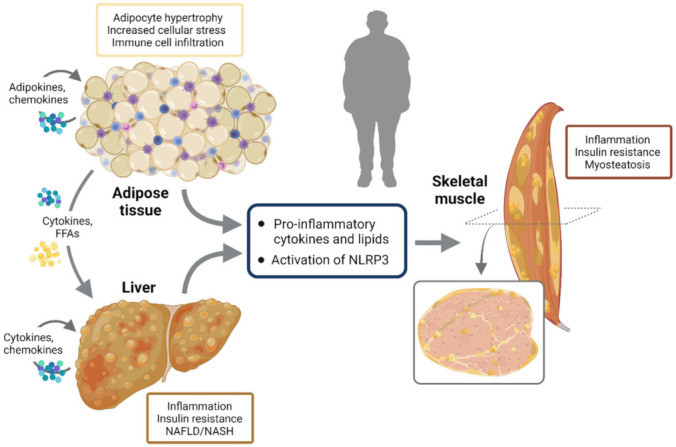
NLRP3 activation and interplay between insulin target tissues. In obesity, adipocyte hypertrophy induces cellular stress together with immune cell infiltration, release of inflammatory adipokines and insulin resistance. Once adipose tissue storage capacity is overwhelmed, ectopic lipid deposit occurs in several other tissues including the liver and skeletal muscle, further worsening inflammation and insulin resistance. This leads to NAFLD/NASH in the liver and to myosteatosis in skeletal muscle. NLRP3 plays a crucial role in metainflammation and in the interplay between these three insulin target tissues.

**Figure 6 cells-10-03023-f006:**
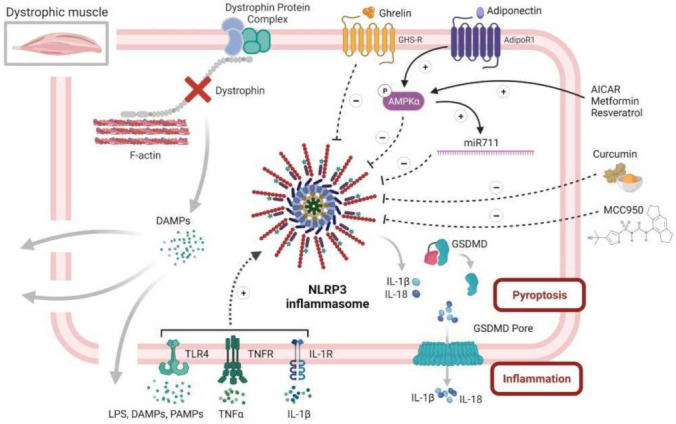
NLRP3 inflammasome and diseased skeletal muscle, case of Duchenne muscular dystrophy. This figure summarizes the central role of NLRP3 in dystrophic muscle inflammation, as well as several of its potential inhibitors. Briefly, the dystrophic muscle is characterized by absence of the dystrophin protein and its associated protein complex, which, in normal cells, links the intracellular cytoskeleton to the extracellular matrix, thus preserving the fiber membrane stability and permeability. Simple contraction of the dystrophic muscle causes microtears in the membrane and the subsequent release of intracellular DAMPS, which, in turn, can act in an autoparacrine manner to activate NLRP3 inflammasome in muscle fibers. Activated NLRP3 leads to an extreme release of inflammatory cytokines and DAMPS, thus maintaining and exaggerating the inflammatory response, eventually leading to pyroptosis and muscle degeneration. Rescuing the dystrophic phenotype can be achieved by alleviating muscle inflammation, in part through repression of NLRP3 inflammasome. This action can be accomplished by several factors. Firstly, adiponectin, a pleiotropic adipokine with potent anti-inflammatory effects, could strongly activate AMPK pathway that represses NLRP3 through reducing NF-κB activity and oxidative stress. Secondly, adiponectin, also through AMPK, increases the expression of its muscle anti-inflammatory mediator, miR-711, which can inhibit both the priming and activation of NLRP3. Thirdly, several drugs and molecules, such as AICAR, resveratrol and metformin, could repress NLRP3 through specific activation of the AMPK pathway. Fourthly, ghrelin, a gastric peptide with anti-inflammatory effect, could also put a brake on muscle inflammation through reduction in NLRP3 activation. Fifthly, curcumin, a potent NF-kB inhibitor, could hinder NLRP3 and the production of inflammatory cytokines. Finally, MCC950, a specific NLRP3 inhibitor, could greatly attenuate the pathogenesis of the dystrophic phenotype, mainly by protecting the muscle from inflammation. Symbols in circles indicate: ‘P’, phosphorylation; ‘+’, activation; and ‘−’, inhibition.

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
