# Peer review of "Walking down Skeletal Muscle Lane: From Inflammasome to Disease"

_cells, 2021, doi:10.3390/cells10113023_

Round 1

Reviewer 1 Report

  The manuscript entitled "Walking down skeletal muscle lane: from inflammasome to disease", which summarizes the function of NLRP3 in diseased skeleton muscles and potential inhibitors targeting NLRP3 to combat muscle disorders. The authors comprehensively conclude NLRP3 inflammasome excessive activation in several skeletal muscle diseases. Further, the article focuses on the inhibitors of NLRP3 and emphasize the ones already tested on muscle related disorders. The manuscript uses several simple figures and cites a considerable number of recent literatures to illustrate the point. Yet, there are corrections and additions that can be made to this manuscript that would serve to improve the way these conclusions are communicated to the readers.

1.Figure 1 ⑤is little confusing, it may be more appropriate to change into inflammatory cytokines release.

2.Please show more information about ionic-mediated pathways for NLRP3 inflammasome activation, which is related to skeletal muscle.

3.Figure 2 does not clearly present the difference between canonical and non-canonical pathway in cleaving pro-IL-1β, pro-IL-18 and Gasdermin D.

4.There are some small grammar mistakes, such as in the line 69 “play “may change into “plays”, line 194 “increasing evidence has” may change into “increasing evidences have”, line 308 the first “As” should be deleted and in the line 619 the “finally” is not proper.

Once the above concerns are fully addressed, the manuscript could be accepted for publication.

Author Response

Dear reviewer, 

Thank you very much for your interesting comments.

1.Figure 1 ⑤ has been modified into "inflammatory cytokines release".

2. Ionic flux including K+ efflux, Ca2+ mobilization, Clefflux, and Na+ influx have been described as NLRP3 inflammasome activator signal. While K+ efflux is considered the most common event for inflammasome activation, the three others ions only play a regulatory role in the inflammatory process.

In the muscle, to our knowledge, there is currently no data concerning Cl and Na+ concentration modification. However, Ca2+ influx and K+ efflux mediated by P2X7R, has been more largely studied. As a result, we clarified the P2X7R action in the "NLRP3 activation" paragraph as followed (line 119-124)

"Several molecular mechanisms have been suggested to lead to NLRP3 inflammasome oligomerization. These include the activation of ATP-dependent P2X purino-receptor 7 receptor (P2 × 7R), which causes both Ca2+ and Nainfluxes thus leading to increased Kefflux via the opening of the two-pore potassium channel 2 (TWIK-2). In addition, the pore formation induced by bacterial toxins, is also inducing increased Kefflux." 

3. In Figure 2, we clarified the difference between canonical and non-canonical pathway by using two different colours (see fig 2).

4.The grammar mistakes, have been adressed as following :

-line 69 “play “ has been changed into “plays”,

-line 194 “increasing evidence has” has been changed into “increasing evidences have”,

-line 308 the first “As” has been deleted

-line 619 the “finally” has been deleted.

Thank you again for your valuable input,

Bw,

Nicolas Dubuisson

Reviewer 2 Report

The Authors present a review entitled “ Walking down skeletal muscle lane: from inflammasome to disease” in which they have explored the pathological roles of inflammasome in the development and progression of numerous diseases, including genetic and acquired myopathies, metabolic diseases and so on. The manuscript is clear and well written. The elucidation of these molecular mechanisms represents the starting point for the identification of  new molecules able to directly act on NLRP3 inflammasome, defining new therapeutic approaches for all these disorders.

Author Response

Dear reviewer,

Thank you very much for your comment and the appreciation of our work.

Best wishes,

Nicolas Dubuisson